# Spatial Analysis of an Emission Inventory from Liquefied Natural Gas Fleet Based on Automatic Identification System Database

**Hoegwon Kim [1], Daisuke Watanabe [2,*], Shigeki Toriumi [3] and Enna Hirata [4]**

[1] Graduate School of Marine Science and Technology, Tokyo University of Marine Science and Technology, Tokyo 135-8533, Japan; navyseal58th@gmail.com

[2] Department of Logistics and Information Engineering, Tokyo University of Marine Science and Technology, Tokyo 135-8533, Japan; daisuke@kaiyodai.ac.jp

[3] Department of Information and System Engineering, Chuo University, Tokyo 112-8551, Japan; toriumi@ise.chuo-u.ac.jp

[4] Center for Mathematical and Data Sciences, Kobe University, Kobe 657-8501, Japan; enna.hirata@platinum.kobe-u.ac.jp

**\*** Correspondence: daisuke@kaiyodai.ac.jp; Tel./Fax: +81-3-5245-7367

**Abstract:** Many states are actively working toward regulating $CO_2$ emissions from a wide range of industries. However, due to the international characteristic of shipping, the emissions from shipping have not yet been strictly controlled. Using Automatic Identification System (AIS) data acquired through satellites, this study estimates the emission inventory, such as, $CO_2$, $CH_4$, $CH_4$, $N_2O$, $NO_x$, CO and non-methane volatile organic compounds (NMVOCs) around the world and bunker consumption from a liquified natural gas (LNG) fleet under the assumption that a LNG fleet uses LNG as fuel. Using position data calculated from an AIS database, we made comparisons regarding the LNG trade amount and bunker consumption of LNG fleet, as well as the total $CO_2$ inventory and $CO_2$ emissions from LNG fleet in the vicinity of the coasts of relevant countries. The result provides insights into (1) how the emissions and bunker consumption from LNG fleet is distributed, (2) which countries are taking relatively more advantages of LNG trade, and (3) which countries are suffering possible harmful effects.

**Keywords:** liquified natural gas (LNG); Automatic Identification System (AIS); spatial analysis; greenhouse gases (GHGs); bunker; emissions

## 1. Introduction

Transportation is the second biggest greenhouse gas (GHG) emission sector, following electric power sector, and most of the emissions come from generating energy using fossil fuels to drive trucks, trains, planes, and vessels [1]. Transportation modes, such as trucks, trains, and planes, are relatively well monitored compared to shipping. However, shipping is the least controlled area. The International Maritime Organization (IMO) has been collecting vessel GHG emission data since 2018 and a long-term plan will be established in 2023 after all the data collected has been analyzed. The European Union (EU) is working on controlling emissions from shipping more actively. The EU decided that shipping would be included in the EU Emission Trading system, which is a market-based measurement based on a cap and trade system, if there is no comparable system operating to control GHG emissions until 2021.

This study aims to obtain insight into emissions from liquified natural gas (LNG) carriers and their relation to countries alongside shipping routes. Section 1 reviews the literature and outlines the background and objectives of the study. Section 2 describes the

Automatic Identification System (AIS) data collection and data imputation. Section 3 calculates vessel emissions. Section 4 aggregates the bunker consumption at the country level and compares it with its LNG trade volumes. Section 5 validates the correlation of the LNG trade volumes and the estimation of the AIS data on a monthly basis. Section 6 summarizes the study.

### 1.1. Background of Study

By carrying a huge amount of cargo in one trip, shipping vessels are known as one of the most eco-friendly modes of transport out of the major transportation modes [2]. In particular, by carrying a huge amount of cargo in one voyage, carrying cargo by shipping vessels is a more efficient means of transportation than other modes of transportation in the aspect of $CO_2$ emissions (tonne-km). Even though the $CO_2$ emissions from shipping are lower than those caused by other means of carrying cargo in terms of tonne-km, the $CO_2$ emissions from shipping reached 1056 million tons in 2018, which represents 2.89% of the worldwide $CO_2$ emissions [3]. The regulations of $CO_2$ emissions in shipping which are currently being implemented were decided by two main organizations. One of these is the IMO and the other is the EU. They came into effect on 1 March 2018, and the first "calendar year" data collection commenced on 1 January 2019. The data collected includes the IMO number; period of calendar year covered; and technical information such as vessel type, gross tonnage, net tonnage, deadweight tonnage, power output, Energy Efficiency Design Index (EEDI) if applicable, ice class, and fuel oil consumption data [4].

In the EU, the Monitoring, Reporting, Verification (MRV) regulations came into effect on 1 July 2015, and they make it mandatory to report and verify $CO_2$ emissions for vessels with over 5000 gross tonnage calling at any EU member state and European Free Trade Association (Norway, Iceland) port. Every year, the responsible party; ship owner; or any other organization or person, such as the manager or bareboat charterer, who has responsibility for the ship operation is required to report the $CO_2$ emissions emitted by the vessel and other required information, including the port of departure and arrival, distance travelled, time spent at sea, amount of cargo carried, and number of passengers [5]. The European Parliament is planning to include shipping in the EU Emission Trading Scheme (ETS), which is basically a cap and trade system, from 2023 if the IMO does not establish a comparable system [6].

### 1.2. Research Review and Objective

Under IMO regulations, Safety of Life At Sea (SOLAS) Chapter V, it is necessary to carry AIS for vessels which have a gross tonnage of over 300 on international transport. The main purpose of the AIS is to avoid collisions at sea. However, the advent of communication technology has made its areas of application wider. Many studies have been carried out on estimating ship inventories using AIS data. Dong et al. [7] systematically reviewed AIS data application in maritime studies and suggested that environmental evaluation is one of the major AIS application fields. Johansson et al. [8] adopted a database of AIS messages for the full year of 2015 for all vessel types and presented a comprehensive global shipping inventory, which can be applied to obtain annual updates of the global ship emissions. Smith et al. [9] implemented a full-scale ship emission inventory analysis using AIS data. Sérgiomabunda et al. [10] estimated a ship emission inventory near the strait of Gibraltar. Coello et al. [11] estimated an emission inventory for the UK fishing fleet. Winther et al. [12] implemented an emission inventory estimation in the artic though a Satellite Automatic Identification System (S-AIS), and Yao et al. [13] estimated ship emission inventories in the estuary of the Yangtze river. The most recent global-scale ship emission inventory analysis was carried out by the IMO Marine Environment Protection Committee (MEPC) [3].

Two major methods with which to derive ship emissions inventories are top-down (fuel-based) and bottom-up (activity-based) [9]. Smith et al. [9] and the IMO MEPC [3] adopted both methodologies, while Jalkanen and Kukkonen [8]; Sérgiomabunda et al.

[10]; Coello et al. [11]; Winther et al. [12]; and Yao et al. [13] adopted the bottom-up methodology. In this study, we adopted the bottom-up methodology to derive fuel consumption.

Power prediction for the ship is one of the most important factors in deriving bunker consumption. Smith et al. [9] and the IMO MEPC [3] used the IHS database; Coello et al. [11] used the statistical fuel consumption; Jalkanen and Kukkonen [8] used the STEAM 3 model; and Winther et al. [12] and Yao et al. [13] adopted the methodology of Kristensen and Lützen [14], which uses the International Towing Tank Conference (ITTC) performance prediction method to obtain the resistance coefficient; and Sérgiomabunda et al. [10] used the ITTC performance prediction method. We adopted the ITTC recommended procedures and guidelines. This allowed us to derive the ship bunker consumption with limited ship specification data. However, the accuracy of the calculation may be improved with comprehensive ship specification data.

The reasons why we chose to analyze the data of LNG carriers are, first, the fact that the demand for gas energy is expected to increase by 1.8% per year from 2015 to 2040. This is much quicker than other conventional modes of energy [15], such as oil (0.6% per year) and coal (0.4% per year). Second, the distribution of the size of LNG carriers is not very wide, which makes it easy to estimate the coefficients related to the calculation of the emissions of LNG carriers. Third, international LNG trade statistics are open to the public, and the import of East Asia countries accounts for more than 60% [16].

The purpose of this study is two-fold. The first is to gain a clear understanding of and insight into the GHGs, such as $CO_2$, $CH_4$, $N_2O$ and other relevant substances, such as $NO_x$, NMVOC, and CO emitted by LNG carriers by visualizing the results of our calculations and the AIS data acquired by satellites. The second is to gain in-depth quantitative insight pertaining to the distribution of the ship emission inventory by applying a geospatial analysis. To visualize and compare the calculated AIS-based bunker consumption and other data, such as the trade of LNG and the total $CO_2$ emissions of each country, data are aggregated through a grid or point and buffer depending on the purpose of each section.

## 2. Automatic Identification System (AIS)

### 2.1. Introduction to AIS

As of 31 December 2004, vessels of over 300 gross tonnage engaged in international voyages and cargo vessels of over 500 gross tonnage not engaged in international voyages are obliged to carry Class A AIS. The motivation for adopting the regulation for carrying AIS is preventing collisions at sea by transmitting vessel data, such as time, position, vessel ID, basic vessel dimensions, and draught. Data are transmitted and received at intervals of 2–10 s while underway and 3 min while anchored. However, the advent of a positioning and communication system broadens the fields of use—AIS data can today be used for purposes such as vessel management, power prediction, and tracking trade flow.

### 2.2. Data Description

The data used in this study were collected by a company named exactEarth. It was founded in 2009 for the purpose of making Satellite AIS data services available to the global maritime market. It currently tracks more than 165,000 vessels through AIS. As exactEarth collects AIS data through satellites, it is possible to obtain AIS data through the ocean regardless of the position of the vessel and regardless of the weather the vessel has faced.

The AIS data used in this study are in the comma-separated values (CSV) format. Every data point is divided by day based on Greenwich Mean Time (GMT). The original data provided by exactEarth include vessel name, callsign, Maritime Mobile Service Identity (MMSI), vessel type, vessel type cargo, vessel class, length, width, flag country, destination, estimated time of arrival (ETA), draught, longitude, latitude, speed over ground

(SOG), course over ground (COG), rate of turn (ROT), heading, navigation (nav) status, source, time, vessel type main, and vessel type sub. The message transmitting interval of AIS is 2–10 s while underway and 3 min at anchor. For the details of vessel type, period, the number of vessels, and the total number of data points used in this study, please see Table 1.

**Table 1.** Outline of data

| Vessel Type | LNG Carrier |
| --- | --- |
| Period | From 2016-01-01 UTC to 2016-06-30 UTC |
| Number of vessels | 327 |
| Total number of data points | 9,072,300 |

Table 2 shows a statistical summary of the data reporting interval of the AIS messages used in this study. The mean reporting interval is about 520 s, and 25% and 75% are 6 and 42 s, respectively. Looking into the sampling rate of the AIS data in more detail, we can see that about 31.7% of the data has a reporting interval of less than 10 s, which is the AIS message transmitting interval for an underway vessel. About 90.6% of the messages have a data reporting interval of less than 3 min, which is same as the AIS message transmitting interval for anchored vessels. About 99.3% of the data has reporting intervals of under 2 h, and 0.7% of the data has reporting intervals greater than 2 h, which seems to be a small number. However, considering that the total number of data points is more than 9 million, the small percentages should not be ignored. The data sampling rate needs to be improved in the future to improve the accuracy of all kinds of AIS-based calculations.

**Table 2.** Distribution of the data interval

| Data Reporting Interval, Hours (A) | Ratio (%) of Data Interval Less than (A) | Number of Data Points with Longer Sampling Rate than (A) |
| --- | --- | --- |
| 2/3600 (2 s) | 7.928 | 8,353,080 |
| 10/3600 (10 s) | 31.656 | 6,200,363 |
| 180/3600 (3 min) | 90.594 | 853,331 |
| 0.5 (30 min) | 95.851 | 376,392 |
| 1 | 97.501 | 226,720 |
| 2 | 99.297 | 63,758 |
| 6 | 99.805 | 17,707 |
| 24 | 99.963 | 3387 |
| 168 | 99.992 | 688 |

Figure 1 shows the distribution of the data samples acquired. South and west of Africa, south of South America, north-east of Australia, and the Indian Ocean are marked as high-concentration areas. Areas such as the East China Sea, the South China Sea, and the Mediterranean Sea are not marked as areas with heavy traffic. This may be because the AIS data collected through satellites show longer data reporting intervals when the vessels are sailing in high-traffic areas compared with low-traffic areas. Few data are observed deviating from the routes of the vessel and on the land side; this might be due to errors that occurred when collecting the data through satellites. In this study, data with this type of error are filtered using the time, position, and speed recorded in the AIS message.

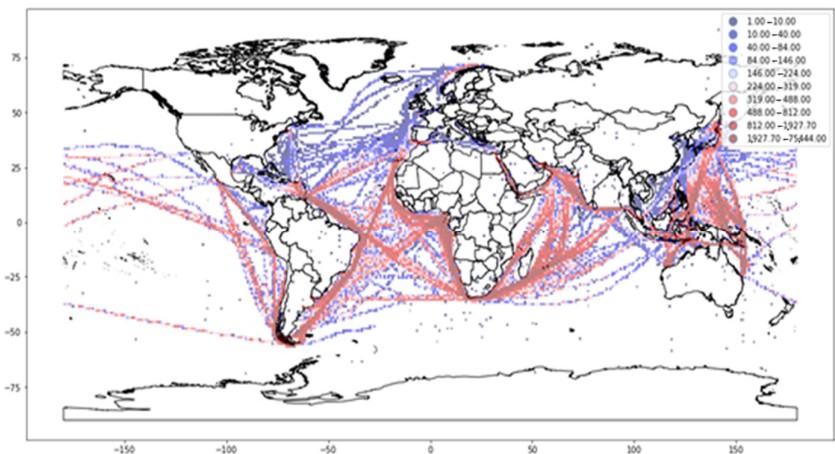

**Figure 1.** Distribution of the data acquired.

Table 3 shows a statistical summary of the ship specification. Most of the vessels are sized from 250 to 300 m in length overall (LOA) and 40–50 m in beam. As the cost of transportation occupies 10% to 30% of the LNG value chain [17], efforts to minimize the cost of transportation may have affected the size of the vessel. For 25% and 75%, the vessel size is 283 m and 291 m, respectively. For the beam, 25% and 75% are 44 m and 48 m, respectively.

**Table 3.** Ship specification

|  | Mean | std | min | 25% | 50% | 75% | Max |
|---|---|---|---|---|---|---|---|
| Length over all (Unit: m) | 284.5 | 35.4 | 69 | 283 | 288 | 291 | 345 |
| Breadth (Unit: m) | 44.8 | 5.6 | 11.8 | 44 | 44 | 48 | 55 |

### 2.3. Imputation

AIS data include many types of information. However, human error or error in communication systems causes problems in terms of data reliability. For example, we found data with a missing MMSI number, an exceptionally high vessel speed, or in a position where a vessel physically cannot pass.

In detail, first, we found that the original data include position data, which shows that a vessel is on the land side or has exceptionally deviated from the route. To remove these data, our plan is to remove data which show a speed higher than a certain knot.

Second, to obtain a reasonable value for speed, we considered the effects of following current, prevailing sea conditions, and intended speed of the vessel, and set the maximum value of the vessel speed as 20 knots.

Third, for the MMIS and ship dimensions, we adopted a vessel tracker and marine traffic which are some of the most famous AIS data providers.

The last item to address is error in ship draught. The data on ship draught in AIS solely relies on the on-board deck officer. It is not rare for the duty officer of a ship to forget to change the value of draught. We analyzed the AIS data and found that the average value of maximum draught–(subtract) minimum draught is about 3.4 m, and about 3.1 m for 75% of the vessels. Taking this into consideration, we replaced the missing draught value with "summer draft–(Subtract) 3",

### 2.4. Origin–Destination Data

As the AIS data included information on the next port of call, it is possible to analyze the origin–destination of the voyages. From the AIS data, origin–destination data are de-

rived. Through the origin–destination trip data and the capacity of the vessel, we calculated the assumed LNG import amount as shown in Table 4. Japan is the biggest LNG import country, followed by unknown (destination country is not identified), Korea, Egypt, Taiwan, China, and India. We will discuss more details of this along with statistical data in Section 3. The reason why "Unknown" ranked second is that, as the destination data in AIS solely rely on the manual input of the onboard officer, errors in data inputting for the destination port can happen. Errors can also happen when data is transferred through a satellite.

**Table 4.** Top 10 assumed LNG import amounts from the AIS origin–destination (port of departure and arrival) data

| No. | Name of Country | Import Amount (Unit: Million Tons) | Percentage |
|---|---|---|---|
| 1 | Japan | 22.74 | 28.44% |
| 2 | Unknown | 22.73 | 28.42% |
| 3 | Korea | 6.95 | 8.69% |
| 4 | Egypt | 4.1 | 5.13% |
| 5 | Taiwan | 3.1 | 3.88% |
| 6 | China | 3.02 | 3.78% |
| 7 | India | 2.93 | 3.66% |
| 8 | Spain | 1.52 | 1.90% |
| 9 | Qatar | 1.01 | 1.26% |
| 10 | United Arab Emirates | 0.91 | 1.14% |
| | Others | 10.96 | 13.71% |
| | Total | 79.97 | 100.00% |

## 3. Vessel Emission Calculation

Figure 2 illustrates the data filtering, ship emission calculation, and visualization process of this section. The data for the LNG fleet are filtered from the original AIS data using the ship type recorded in the AIS message. We also remove the messages with the wrong position using the time, position, and speed recorded in the AIS data. Then, using the vessel dimensions, speed, and position data included in the AIS message, we calculate the total resistance when the vessel is sailing at speed V, following the method included in the International Towing Tank Conference (ITTC) recommended procedure [18]. As the vessel performance could vary depending on the condition of the hull, the weather, the current, etc., a margin of error should be considered when calculating the power requirement. From the calculated power, using the specific fuel oil consumption (SFOC) and emission factor, a calculation can be performed to obtain the bunker consumption and emission inventory. Python (version 3.6.6) was adopted as the data manipulation language. We adopted the Python module Pandas (version 0.23.4) to aggregate the calculated emissions and QGIS (version 2.14.21) for visualization.

Several studies have been carried out on the estimation of vessel resistance, which is key to calculating the required power and bunker consumption when the vessel is sailing at specific speed *V*. In this study, we adopted the method recommended by the ITTC [18] to estimate the total resistance, which is key to derive the power requirement. The detailed parameters used in this study are taken from international organizations and public sources (i.e., Takahashi et al. [19]; Kristensen and Lützen [14]).

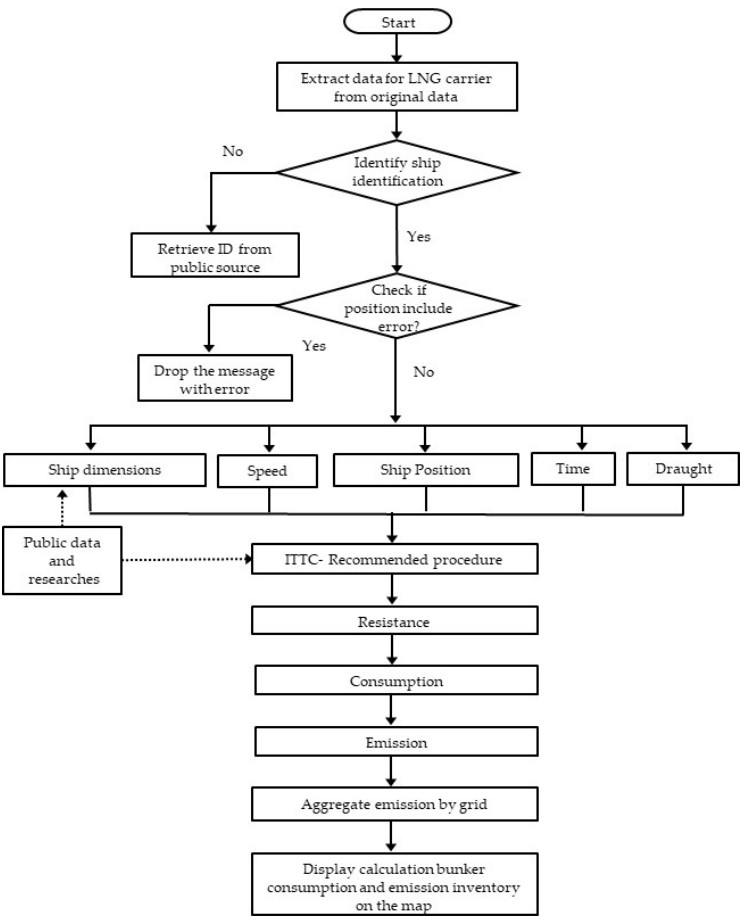

**Figure 2.** Flowchart of data manipulation in Section 3.

*3.1. Total Resistance*

When calculating, the total resistance draught, T, and speed, V, which are included in the AIS data, are used as variables. The speed and draught change over time due to external forces and how much cargo and bunker are on board the ship. In order to calculate the power required when the vessel is sailing at speed V, it is necessary to derive the total resistance first. The total resistance can be denoted as [18]

$$R_T = \frac{1}{2} \times C_T \times \rho \times S \times V^2, \tag{1}$$

where $R_T$ is the total resistance, $C_T$ is the total resistance coefficient, $\rho$ is the density of water, $S$ is the wetted surface of the hull, and $V$ is the speed of the vessel. $C_T$, the total resistance coefficient, can be derived from

$$C_T = C_F \times C_A \times C_{AA} \times C_R, \tag{2}$$

where $C_F$ is the frictional resistance coefficient, $C_A$ is the incremental resistance coefficient, $C_{AA}$ is the air resistance coefficient, $C_R$ is the residual resistance coefficient. The $C_F$ of the hull often causes some 70–90% of the vessel's total resistance for a low-speed vessel (bulk carriers and tankers), and sometimes less than 40% of the vessel's total resistance for a high-speed vessel [20]. $C_F$ can be described as [18]

$$C_F = \frac{0.075}{(log_{10}R_n - 2)^2}, \tag{3}$$

where $R_n$ is the Reynolds number, which is described as

$$R_n = \frac{V * L_{WL}}{\varphi}, \tag{4}$$

where $L_{WL}$ is length of the waterline and $\varphi$ is the kinematic viscosity of the water. In this study, the value for $\varphi$ is adopted from the study conducted by Lienhard [21].

$C_F$, the frictional resistance coefficient, concerns the roughness of the hull surface. As the surface roughness of the model is different from the roughness of the vessel, when calculating the resistance coefficient an incremental resistance coefficient, $C_A$, is added. The value of $C_A$ can be estimated using the following expression [14]

$$1000 * C_A = Max\left\{-0.1; \ 0.5 \times log(\Delta) - 0.1 \times \left(log(\Delta)\right)^2\right\}, \tag{5}$$

where $\Delta$ is the displacement of the vessel, which can be denoted as

$$\Delta = C_B \times L_{PP} \times B \times T, \tag{6}$$

where $C_B$ is the block coefficient of the vessel, $L_{PP}$ is the length between perpendiculars, $B$ is the beam of the vessel, and $T$ is the draught of the vessel.

The value for $C_{AA}$ is derived from the study carried out by Kristensen et al. [14]. The value for $C_R$ is adopted from the study implemented by Kristensen et al. [14]. Finally, the wetted surface, S, for tankers and bulk carriers can be derived by [14]

$$S = 0.99 \times \left(\frac{\Delta}{T} + 1.9 \times L_{WL} \times T\right). \tag{7}$$

*3.2. Power Prediction*

Based on the calculated total resistance of the vessel, the required power when the vessel is sailing at speed V in calm sea conditions can be calculated by considering the components of the propulsion efficiencies. The installed power is the power required to tow a vessel with speed V in a calm sea. The installed power can be derived from [22]

$$P_I = \frac{R_T \times V}{(\eta_D \times \eta_T)} + m, \tag{8}$$

where $P_I$ is the installed power, $\eta_T$ is the transmission efficiency, $\eta_D$ is the quasi-propulsive coefficient, and $m$ is the sea margin.

*3.3. Bunker Consumption and Emission Pollutants*

The bunker consumption can be derived by multiplying $P_I$ by the SFOC in Table A1 [9] in Appendix A. The calculated bunker consumption amount is 3,540,342.2 tons. To calculate how much emission pollutants are released from the LNG fleet, we adopted the emission factors introduced by Smith et al. [9]. The amount of emission pollutants can be derived by multiplying the bunker consumption by the emission factors in Table A2 [9] in Appendix A. The calculated emission inventory is shown in Table 5.

To achieve a deeper insight into the distribution of bunker consumption, we plotted the result on a map (Figure 3). Highly concentrated routes are mostly located from the Middle East to the Far East (Arabian Sea–Indian Ocean–Malacca Strait–Singapore Strait–South/East China Sea–West Pacific) and the Middle East to Europe (Arabian Sea–Red Sea–Suez–Mediterranean Sea), and Oceania to the Far East (Indonesian Archipelago–South/East China Sea–West Pacific). As the emission inventory is derived from the product of bunker consumption and the emission factor, the distribution of each air pollutant is the same as in Figure 3.

**Table 5.** Emission inventory of the LNG fleet

| Emission Pollutant | Amount (Metric Tons) |
|---|---|
| $CO_2$ | 9,735,941.05 |
| $CH_4$ | 181,265.52 |
| $N_2O$ | 389.44 |
| $NO_x$ | 27,720.88 |
| CO | 27,720.88 |
| NMVOC | 10,656.43 |

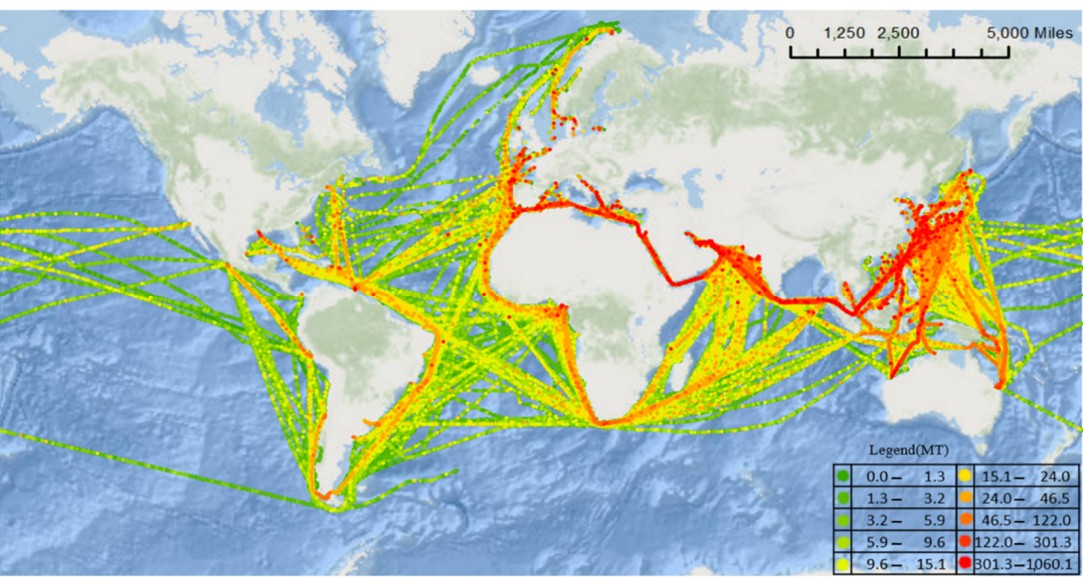

**Figure 3.** Distribution of bunker consumption by LNG fleet, 1*1 degree.

Compared to Figure 1 (distribution of number of data acquired), Figure 3 shows a high density in the Mediterranean Sea, south of the Bay of Bengal, in the Malacca Strait, in the South China Sea, in the East China Sea, and along the coast of Japan and Korea. This might be the factor supporting the point that a higher number of data points collected does not mean more vessel activity in the area for a specific ship type.

## 4. Comparison between Bunker Consumption, LNG Trade Amount, and $CO_2$ Emission from LNG Fleet at the Vicinity of Each Country

The flow of this chapter uses the emission data calculated in Section 3, the position recorded in the AIS message, and the global country boundary data. First, we aggregated the bunker consumption from 0.2 degrees from the coast of each country. Geopandas (version 0.4.0) was used for the tool for buffering and aggregation in this section. Second, from the aggregated data by country, we made a comparison with the international trade data [16] to gain a clearer understanding about which countries are taking advantage of LNG trade and which countries are suffering from the unfavorable effects from the trade of LNG. In addition to this, from the aggregated bunker consumption we calculated the $CO_2$ emissions and compare them with the entire $CO_2$ inventory of each country.

### 4.1. Buffer

To aggregate the bunker consumption in the vicinity of coast of each country, we adopted a buffer. Buffers are areas around the point, line, polygon, or group of it. For example, buffering a point returns a round shape area and buffering a line returns a lane shape area. A buffer could be a great analysis tool. For example, same as what we did in

this study, it can create the area from fixed distance (0.2 degree) away from the coast of each country. The reason why we adopted 0.2 degrees in this study is that 0.2 degrees is 12 min which means 12 nautical miles in equator. In UN convention on the law of the sea part 2 "Territorial Sea And Contiguous Zone", Section 2 "Limits Of The Territorial Sea", article 2 states that every state has the right to establish the breadth of its territorial sea up to a limit not exceeding 12 nautical miles, measured from the baselines determined in accordance with this convention. Data for coastline of each country are obtained from the Environmental Systems Research Institute (ESRI). As every calculated bunker consumption has a position, we aggregated the bunker consumption inside of the buffer.

### 4.2. Result

### 4.2.1. Comparison Bunker Consumption near the Coastline of Each Country and LNG Trade Amount

The left side of Table 6 illustrates details of how much bunker consumption made inside of buffer created. The counties listed on this table are not only located at the end of the route but also located along the main passage of transportation.

The right side of Table 6 illustrates the summation of the LNG export and import of each country LNG with the export and import data from the IGU World LNG report 2017 [16]. Few countries listed on left side of Table 6 are not listed in right side of Table 6, and the order of list is quite different. This may give a clearer understanding of which countries are actively involved in LNG trade and which countries may be affected by the emissions of the LNG fleet. Especially, countries such as Sri Lanka and Djibouti are not actively involved in the trade of LNG; however, those countries have a high possibility of being affected by the air pollutants emitted from the LNG fleet.

**Table 6.** Bunker consumption made from 0.2 degrees from each country (top 20) and the sum of LNG export and import for the top 20 countries (million tons per annum) [16]

| No. | Country | Bunker Consumption (Metric Tons) | No. | Country | Sum of LNG Export and Import (Million Tons) |
|---|---|---|---|---|---|
| 1 | Malaysia | 69,193.20 | 1 | Japan | 83.34 |
| 2 | Indonesia | 42,643.16 | 2 | Qatar | 77.24 |
| 3 | Egypt | 38,887.04 | 3 | Malaysia | 51.07 |
| 4 | Japan | 33,839.04 | 4 | Australia | 44.34 |
| 5 | Yemen | 25,016.19 | 5 | Korea | 33.71 |
| 6 | Iran | 23,185.27 | 6 | China | 26.78 |
| 7 | Singapore | 18,171.99 | 7 | India | 19.17 |
| 8 | Oman | 15,807.31 | 8 | Nigeria | 18.57 |
| 9 | Qatar | 9291.88 | 9 | Indonesia | 16.59 |
| 10 | Philippines | 7625.87 | 10 | Taiwan | 15.04 |
| 11 | Papua New Guinea | 6222.00 | 11 | United Arab Emirates | 14.09 |
| 12 | Spain | 5561.60 | 12 | Algeria | 11.52 |
| 13 | India | 5219.54 | 13 | Russia | 10.84 |
| 14 | Australia | 4822.99 | 14 | Trinidad and Tobago | 10.57 |
| 15 | Chile | 4065.93 | 15 | Spain | 9.88 |
| 16 | Trinidad and Tobago | 3358.28 | 16 | Oman | 8.14 |
| 17 | Korea | 3335.94 | 17 | Egypt | 7.83 |
| 18 | Greece | 2954.19 | 18 | United States | 7.44 |
| 19 | Djibouti | 2542.90 | 19 | United Kingdom | 7.37 |
| 20 | Morocco | 2444.64 | 20 | Papua New Guinea | 7.36 |

Figure 4 is the scatter plot of the bunker consumption aggregated inside of the buffer created and the LNG trade amount of each county. It gives a quick insight into which counties are benefiting more from the LNG trade. Many countries are enjoying the advantages of international shipping; however, countries—including Malaysia, Indonesia, Yemen, Oman, Philippines, Sri Lanka, Chile, etc.—who share the coast with a main passage of shipping (e.g., Strait of Malacca, Indonesian Archipelago, Arabian Sea, Mediterranean Sea, Magellan Strait, etc.) may not gain enough benefit from international shipping.

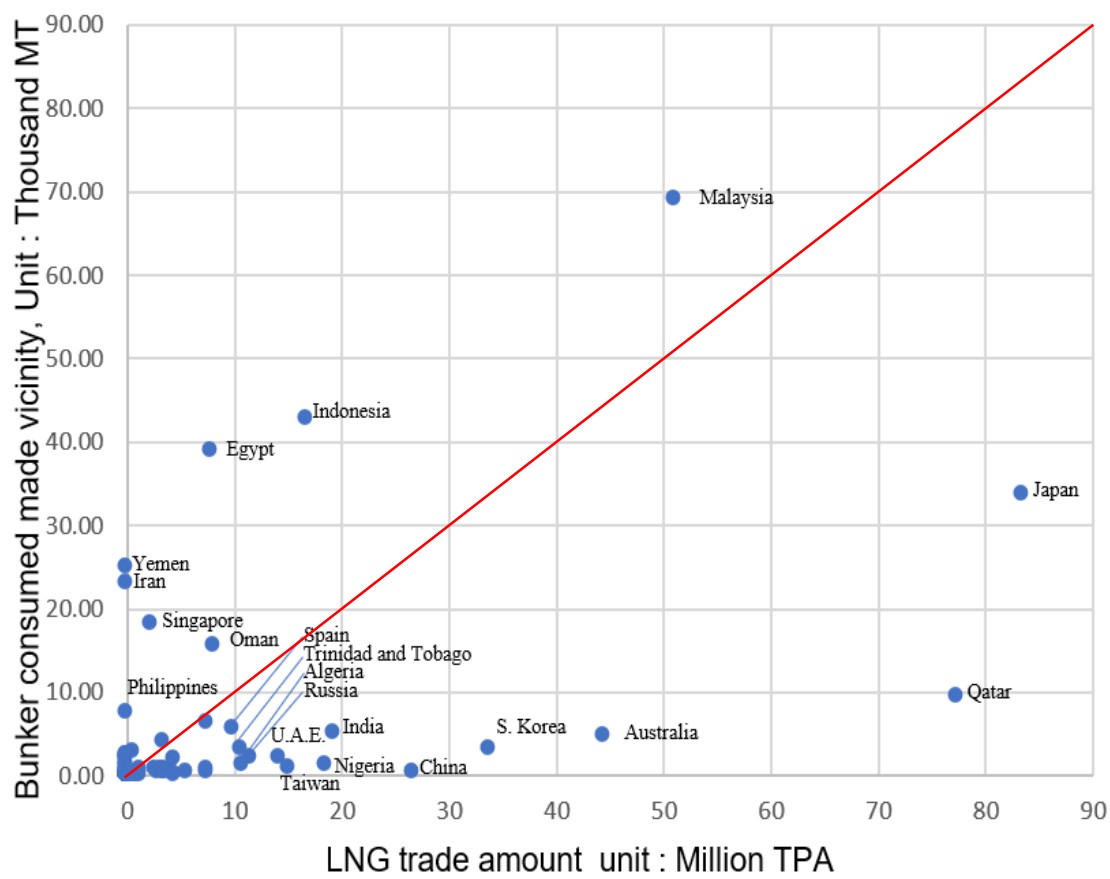

**Figure 4.** Scatter plot of the LNG trade amount and the sum of emissions made 0.2 degrees away from the coast.

### 4.2.2. Comparison of the $CO_2$ Emissions from the LNG Fleet and the $CO_2$ Inventory of Each Country

We compared the $CO_2$ emission amount from LNG fleet calculated in this study with $CO_2$ emission of each country sourced from Emission Database for Global Atmospheric Research [23]. Table 7 shows ratio of $CO_2$ emission from LNG compared to $CO_2$ emission of each country. Countries such as Malaysia, Timor-Leste, Yemen, Papua New Guinea, Oman, Egypt, Indonesia, and Sri Lanka are relatively more affected by emission from LNG fleet in country scale. Table 7 outlines detailed value of percentage.

**Table 7.** Comparison of the calculated $CO_2$ emissions and entire $CO_2$ emissions by country (top 20 sorted by $CO_2$ from LNG fleet ‰)

| | Calculated Bunker Consumption (Unit: ton) | Yearly $CO_2$ Emission of Whole Country (Unit: Kilo-ton) | Half of Yearly $CO_2$ Emission (*A*) (Unit: ton) | Calculated $CO_2$ Emission Amount (*B*) (Unit: ton) | $CO_2$ from LNG Fleet (*B/A*) (Unit: ‰) |
|---|---|---|---|---|---|
| Eritrea | 2347 | 684 | 342,070 | 6455 | 188.70 |
| Timor-Leste | 1653 | 496 | 247,845 | 4546 | 183.42 |
| Djibouti | 2543 | 1509 | 754,425 | 6993 | 92.69 |
| Gibraltar | 807 | 573 | 286,355 | 2219 | 77.49 |
| Puerto Rico | 720 | 713 | 356,380 | 1980 | 55.56 |
| Yemen | 25,016 | 25,648 | 12,823,995 | 68,795 | 53.65 |
| Comoros | 105 | 108 | 54,210 | 289 | 53.31 |
| Papua New Guinea | 6222 | 9087 | 4,543,495 | 17,110 | 37.66 |
| Sao Tome and Principe | 27 | 56 | 28,090 | 74 | 26.34 |
| Saint Helena | 6 | 13.13 | 6565 | 16 | 24.37 |
| Singapore | 18,172 | 48,382 | 24,190,880 | 49,973 | 20.66 |
| Malaysia | 69,193 | 266,252 | 133,125,770 | 190,281 | 14.29 |
| Equatorial Guinea | 507 | 2156 | 1,078,185 | 1393 | 12.92 |
| Anguilla | 6 | 30 | 15,130 | 17 | 11.24 |
| Oman | 15,807 | 87,836 | 43,917,885 | 43,470 | 9.90 |
| Egypt | 38,887 | 219,377 | 109,688,675 | 106,939 | 9.75 |
| Trinidad and Tobago | 3358 | 34,974 | 17,487,130 | 9235 | 5.28 |
| Qatar | 9292 | 98,990 | 49,495,040 | 25,553 | 5.16 |
| Mauritius | 283 | 3192 | 1,596,155 | 779 | 4.88 |
| Indonesia | 42,643 | 530,036 | 265,017,825 | 117,269 | 4.42 |

## 5. Validation

To verify that the calculated bunker consumption and air pollutant amount can be explained, first, we re-arranged the daily sum of bunker consumption. From the daily sum of bunker consumption, we made a series of data which is the sum of few days including that day—i.e., data for 4 January in the sum of three days means the sum of bunker consumption from 2 to 4 January. As the period of summation for each data increases, the difference between the mean and the median decreases and the increase in standard deviation is relatively smaller than that of the mean of data. Table 8 shows the detail of the validation. As Japan and Korea were the world's biggest and second-biggest LNG importer [16], we adopted LNG import statics of Japan published by the Japanese Ministry of Economy, Trade, and Industry [24] and data published by the Korea Gas Corporation (KOGAS) was adopted. As the monthly import amount of LNG was not included in the data released by KOGAS, we used the monthly number of voyages for Korea [25]. The Korea Gas Corporation is a state-owned company and accounted for about 90% of the entire LNG import of Korea in 2016 [26,27].

**Table 8.** Moving average (MA) of consumptions (unit: thousand MT)

| | Daily Sum | 3-Day MA | 7-Day MA | 14-Day MA | 28-Day MA | 56-Day MA | 84-Day MA |
|---|---|---|---|---|---|---|---|
| Mean | 19.45 | 58.60 | 137.46 | 275.95 | 549.98 | 1092.82 | 1632.56 |
| STD | 12.45 | 20.61 | 24.24 | 26.08 | 33.96 | 39.60 | 42.47 |
| Median | 15.37 | 49.37 | 143.36 | 275.35 | 551.46 | 1090.39 | 1633.07 |

Figure 5 shows the LNG import amount in Japan [24] and the number of voyages of LNG fleet in Korea [25] in 2016. The LNG import in Japan in March recorded the highest import amount, followed by February and January in 2016. The number of voyages of LNG carriers which transported LNG to South Korea in December the recorded highest number of voyages, followed by January and March in 2016. For both the LNG import amount of Japan and the number of voyages of South Korea, January to March are noticeably higher than the values recorded from April to June 2016.

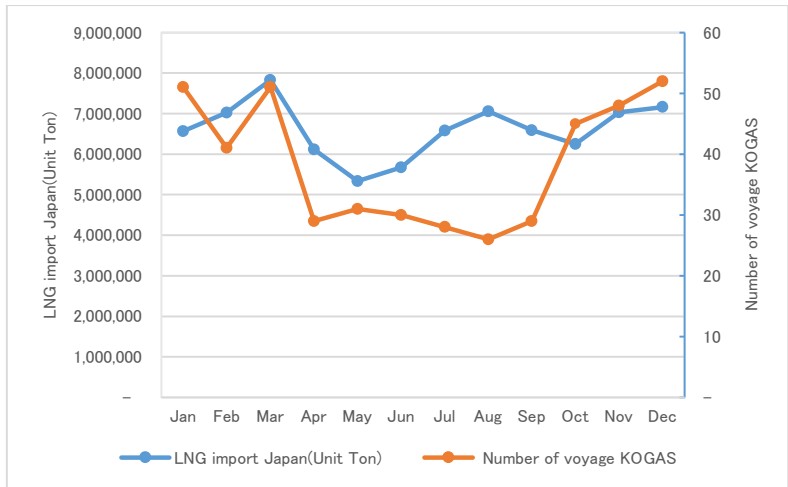

**Figure 5.** LNG import amount in Japan and e number of voyages of LNG fleet in Korea in 2016.

Looking into detail at the Japanese statistical data of the LNG trade volume and the LNG trade volume assumed from the AIS data, Table 9 shows monthly comparison of the statistical data released by the Ministry of Economy, Trade, and Industry (METI) of Japan and AIS based estimated LNG trade volume of Japan. The ratio estimated from the AIS data is about 59% compared to the data released by METI. Depending on the month, the percentage of estimated trade volume from AIS, compared to the data from METI, varied from 47% to 83%.

**Table 9.** Monthly comparison of the statistical data and estimated LNG trade volume from AIS, Japan.

| Month | Estimated from AIS (*A*) (Unit: MT) | Statistics from METI (*B*) (Unit: MT) | $\frac{A}{B}$ |
|---|---|---|---|
| Jan-16 | 3.07 | 6.57 | 0.47 |
| Feb-16 | 4.48 | 7.02 | 0.64 |
| Mar-16 | 4.06 | 7.83 | 0.52 |
| Apr-16 | 3.84 | 6.11 | 0.63 |
| May-16 | 2.56 | 5.34 | 0.48 |
| Jun-16 | 4.73 | 5.67 | 0.83 |
| Sum | 22.74 | 38.54 | 0.59 |

In the case of Korea, we calculated the monthly amount of import from the KOGAS data as shown in Table 10. From the total number of voyages (461 voyages) [26,27] and total import amount (31,846,875 tons) [26,27] in 2016, we derived the average amount of LNG carried per voyage (D in Table 10). Then, by the multiple number of voyages (N in of Table 10) with D, the monthly amount imported by KOGAS is calculated. As KOGAS imported about 92.5% [26,27] of the total LNG import amount of Korea, by dividing B (Table 10) by 0.925 we calculated the monthly import amount of Korea (C in Table 10). Depending on the month, 34–58% of the LNG trade volume was covered by the AIS origin–destination data. In both Japan and Korea's case, the % of estimated amount from AIS was the highest in June (Tables 9 and 10).

**Table 10.** Monthly comparison of the statistical data and estimated LNG trade volume from AIS, Korea

| Month | Estimated from AIS (*A*) (Unit: Million Tons) | Number of Voyages (*N*) | Average Carried Amount per Voyage (*D*) (Unit: ton) | KOGAS (*B* = *N*D*) (Unit: Million Tons) | Extrapolated Import Amount (*C* = *B*/0.925) (Unit: Million Tons) | $\frac{A}{C}$ |
|---|---|---|---|---|---|---|
| Jan-16 | 1.29 | 51 | | 3.52 | 3.81 | 0.34 |
| Feb-16 | 1.48 | 41 | | 2.83 | 3.06 | 0.48 |
| Mar-16 | 1.43 | 51 | | 3.52 | 3.81 | 0.38 |
| Apr-16 | 0.89 | 29 | 69,082.16 | 2.00 | 2.17 | 0.41 |
| May-16 | 0.55 | 31 | | 2.14 | 2.32 | 0.24 |
| Jun-16 | 1.30 | 30 | | 2.07 | 2.24 | 0.58 |
| Sum | 6.94 | 233 | 69,082.16 | 16.08 | 17.41 | 0.40 |

The reasons why the estimated trade volume based on the AIS data is smaller than that of the statistical data might be errors in the destination country, errors in classifying the loading conditions, errors in method used to separate voyages, errors in the loading capacity of the vessel, or incomplete destination databases and tracking.

Figure 6 shows the correlation between the MA of bunker consumption and the LNG import amount of Japan (orange line) [24], and the number of voyages of Korea (blue line) [25]. As the period of summation increases, both correlations show a similar trend. Especially, the correlation coefficient with the 56-day MA, 8 weeks, is higher than 0.8. This may imply that the LNG fleet movement is related to the planned LNG importing amount 1–2 months later.

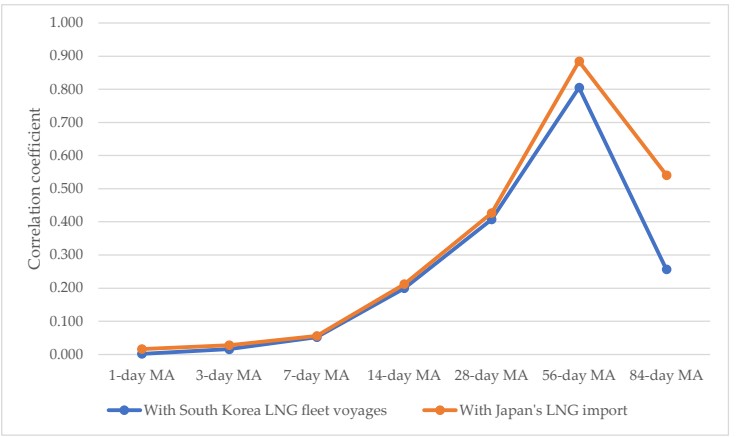

**Figure 6.** Correlation coefficient between the sum of bunker consumption and LNG import amount and the sum of bunker consumption and the number of voyages.

In conclusion, the bunker consumption of LNG fleet is correlated to the LNG trade volume in the case of Japan and Korea, the top 2 largest LNG importers.

## 6. Summary

This paper aims to offer insight into LNG emission inventory and provides empirical evidence for the finding that some countries who do not benefit from LNG trade suffer high emissions near their coasts. The contribution of this paper is four-fold. First, we estimated vessel resistance in accordance with the ITTC recommended procedure and derived the bunker consumption and emission inventory based on AIS data. Second, by plotting it on a map, we obtained a deeper understanding of emissions and bunker consumption. Third, we applied a geospatial analysis to ship emission inventories to figure out how the air pollutant emissions and bunker consumption distributions are clustered. Fourth, by calculating the sum of the bunker consumption, which can be easily converted

to emission inventory from the coast of each country, we were able to illustrate how much each country might be affected by emissions from the LNG fleet.

The research result also offers managerial implications. Ships emit along the main LNG shipping routes, such as the Strait of Malacca, the Indonesian Archipelago, the Arabian Sea, the Singapore Strait, the Mediterranean Sea, the Magellan Strait, etc. Many countries, such as Sri Lanka and the Philippines, located in the vicinity of these routes are not actively involved in the trade of LNG or are unable to enjoy much of the prosperity from shipping, but have a high amount of bunker consumption near the coast of their country. By comparing the amount of bunker consumption 0.2 degrees away from the coast of each country with international LNG trade and the amount of $CO_2$ emitted 0.2 degrees away from the coast of each country, we gained an understanding of which counties are taking relatively more advantage of LNG trade and which countries are suffering relatively more from the probable harmful effects. International society may need to think about how it can compensate these countries for the possible damage from ship-emitted pollutants.

This research could be improved in many ways. First, the accuracy of calculation could be improved by resolving the problem of unstable AIS data intervals. Second, with more detailed data on the fuel efficiency and engine type of LNG carriers, more accurate results could be gained. Third, with a more extensive amount of AIS data points, seasonal and monthly trends could be analyzed. Finally, same approaches could be applied to other types of vessel to gain a more extensive understanding of ship emissions.

**Author Contributions:** Conceptualization, W.H.; Methodology, K.H.; Software, K.H.; Validation, W.H., T.S. and H.E.; Writing—original draft preparation, K.H., W.H., T.S. and H.E.; Writing—review and editing, K.H., W.H. and H.E. Funding, W.H. and T.S. All authors have read and agreed to the published version of the manuscript.

**Funding:** This research was funded by Japan Society for the Promotion of Science (JSPS KAKENHI), grant numbers 18K04601 and 17H02042.

**Institutional Review Board Statement:** Not applicable.

**Informed Consent Statement:** Not applicable.

**Conflicts of Interest:** The authors declare no conflict of interest.

### Appendix A

Table A1 shows SFOC by engine age [9] and distribution of engine age of the vessels included in AIS data. SFOC is commonly expressed in g/kW·h. As the vessel engine gets older, an efficiency of the engine goes down and advent of technology make a newer engine more efficient.

**Table A1.** SFOC [9] and the distribution of the engine age of the vessels included in the AIS data

| Engine Age | MSD (Medium-Speed Diesel (Engine)), Unit: g/kW·h. | Number of Vessels by Engine Age, Unit: Year | Percentage |
| --- | --- | --- | --- |
| Before 1983 | 215 | 6 | 1.83% |
| 1984–2000 | 195 | 46 | 14.07% |
| After 2001 | 185 | 275 | 84.10% |

Table A2 shows emission factors for top-down emissions from combustion of fuels. Using emission factors, amount of emitted air pollutant could be derived from bunker consumption amount. It is expressed in kg/kg of fuel.

**Table A2.** Emission factors of each emission pollutants [9]

| Emission Pollutant | Emission Factor (kg/kg of Fuel) |
|---|---|
| $CO_2$ | 2.75000 |
| $CH_4$ | 0.05120 |
| $N_2O$ | 0.00011 |
| $NO_x$ | 0.00783 |
| CO | 0.00783 |
| NMVOC | 0.00301 |

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
