# Peer review of "Spatial Analysis of an Emission Inventory from Liquefied Natural Gas Fleet Based on Automatic Identification System Database"

_sustainability, doi:10.3390/su13031250_

Round 1
Reviewer 1 Report
The paper undertakes relatively rare publication area of emission of CO2 from shipping. Insight on emission from LNG carriers which take into the consideration different countires losses seem to be valuable approach which on the basement of original methodology is searching for important environmental information. Specially sophisticated and rebuilt methodology and data processing deserves to be emphasized as committed to making contribution.
However, the paper requires minor corrections, such as:
- a large amount of data presented in a descriptive form introduces unnecessary difficulties in understanding the authors' intentions and gives the impression of chaos; some of the data can be presented in a more structured way in tabular form, some of the data can also be more organized at the description stage,
- in the „Background of study” Authors cited data from 2012 about CO2 emission, which from the perspective the rest od the paper is too old data; as such information is wide available, more relevant nowadays data should be presented there,
- Table 1 title is omitted,
- there are many language mistakes and typing errors, np. under Table 2 in the sentence „It may because AIS data…”lack of the verb etc., in 3.1. „The When…” and others,
- conclusions should be broadened to identify possible more concrete benefits for selected countries, especially the most disadvantaged; a large amount of research effort put by the authors may have a chance of interest from various representatives of these countries, but the lack of an explicitly indicated added value may be an unnecessary limitation.
Author Response
Dear reviewer,
Thank you for the thoughtful and constructive feedback you provided regarding our manuscript.
We have updated our manuscript based on your advice to make improvements and the check through English language editing.
(1) - a large amount of data presented in a descriptive form introduces unnecessary difficulties in understanding the authors' intentions and gives the impression of chaos; some of the data can be presented in a more structured way in tabular form, some of the data can also be more organized at the description stage,
-> We have added table 3, 8, A1, A2 to add to what you have been written. In addition to it, we have fixed table format.
(2) - in the „Background of study” Authors cited data from 2012 about CO2 emission, which from the perspective the rest od the paper is too old data; as such information is wide available, more relevant nowadays data should be presented there,
-> We have added fourth IMO study which includes CO2 emission from shipping from 2012 to 2018. (Released on 28 July 2020).
(3) - Table 1 title is omitted,
-> Title for table 1 have added (Title : Outline of data)
(4)- there are many language mistakes and typing errors, np. under Table 2 in the sentence „It may because AIS data…”lack of the verb etc., in 3.1. „The When…” and others,
-> We have got English language editing service to fix the formal mistakes.
(5) - conclusions should be broadened to identify possible more concrete benefits for selected countries, especially the most disadvantaged; a large amount of research effort put by the authors may have a chance of interest from various representatives of these countries, but the lack of an explicitly indicated added value may be an unnecessary limitation.
-> In this study, what we have intended was illustrate current status in first place. We would like to investigate ideas and solutions what kind of action should be taken by international organizations and governments in next research.
With these changes to our final manuscript, we hereby resubmit our manuscript for a secondary evaluation. Thank you once again for your consideration of our paper.
Sincerely,

Reviewer 2 Report
This paper aims to offer insights about LNG emission and tries to provide empirical evidence for the finding that some countries who are not benefiting from LNG trade are suffering high emissions near their coasts.
When the paper estimates vessel resistance (formulae 1 and following), it lacks contextualization with respect to previous and present theoretical background. Authors should provide it. Furthermore, there are clear formal mistakes to be corrected. E.g. what is read in line 206 as "The When calculate". It makes no sense.
The same can be said in line 128 ("colleting interval of less than 2 hours"), or when using misplaced formulae in line 236 or when using expressions with no definite sense as "by calculating sum of bunker consumption". A general review of English expressions should be provided to clarify the paper.
Author Response
Dear reviewer,
Thank you for the thoughtful and constructive feedback you provided regarding our manuscript.
We have updated our manuscript based on your advice to make improvements and the check through English language editing.
1. When the paper estimates vessel resistance (formulae 1 and following), it lacks contextualization with respect to previous and present theoretical background. Authors should provide it.
-> We have updated "chapter 1.1 research review and objective" to add to what you have been written.
2. Furthermore, there are clear formal mistakes to be corrected. E.g. what is read in line 206 as "The When calculate". It makes no sense. The same can be said in line 128 ("colleting interval of less than 2 hours"), or when using misplaced formulae in line 236 or when using expressions with no definite sense as "by calculating sum of bunker consumption". A general review of English expressions should be provided to clarify the paper.
-> We have got English language editing to fix the formal mistakes.
With these changes to our final manuscript, we hereby resubmit our manuscript for a secondary evaluation. Thank you once again for your consideration of our paper.
Sincerely,

Reviewer 3 Report
In current manuscript, author have carried out a spatial analysis of automatic identification system based LNG fleet emission inventory worldwide. The topic is interesting, but this manuscript could be improved by means of reviews as recommended below.
I suggest authors introduce following changes before this manuscript can be considered for publication.
- Authors could avoid using abbreviations in the title of the manuscript. Well-known abbreviations would fit better in keywords.
- Authors could include in the abstract a small explanation of the estimated emissions (for example: "this study estimates emission inventory, namely, CO2, CH4, N2O, NOx, CO and NMVOC around the world").
- Abbreviations should be defined the first time they are written. However, LNG was not defined.
- Introduction section mentions GHGs but does not explain which gases were analyzed and why (perhaps because they are the most important associated with the analyzed sector).
- Figures 1 and 3 could improve the quality. In addition, it is recommended to send a higher resolution images that can be zoomed into the web version of the article when it is published.
- In table 3, authors could add an additional column that shows “%” of Import amount Million tons in relation to the total.
- It is highly recommended to add a table as an annex that shows the main emission factors used, as well as other values considered in section 3 (Vessel emission calculation).
Author Response
Dear reviewer,
Thank you for the thoughtful and constructive feedback you provided regarding our manuscript.
We have updated our manuscript based on your advice to make improvements and the check through English language editing.
1.Authors could avoid using abbreviations in the title of the manuscript. Well-known abbreviations would fit better in keywords.
→ Title have been updated. (New title : Spatial analysis of an emission inventory from liquefied natural gas fleet based on automatic identification system database)
2.Authors could include in the abstract a small explanation of the estimated emissions (for example: "this study estimates emission inventory, namely, CO2, CH4, N2O, NOx, CO and NMVOC around the world").
→ We have added what emission materials are derived in this study. Please find abstract of updated manuscript.
3.Abbreviations should be defined the first time they are written. However, LNG was not defined.
→ We have defined LNG and have thoroughly investigated manuscript again.
4. Introduction section mentions GHGs but does not explain which gases were analyzed and why (perhaps because they are the most important associated with the analyzed sector).
→We have added what you written. Please find line 109-117 of updated manuscript.
5. Figures 1 and 3 could improve the quality. In addition, it is recommended to send a higher resolution images that can be zoomed into the web version of the article when it is published.
→ Figure 3 has been replaced to new image with a higher resolution as it is one of the most important results of our research. However, we have not updated figure 1 because it is just showing density of data by area.
6. In table 3, authors could add an additional column that shows “%” of Import amount Million tons in relation to the total.
→ We have added import ratio column. Please find table 4 on updated manuscript.
7.It is highly recommended to add a table as an annex that shows the main emission factors used, as well as other values considered in section 3 (Vessel emission calculation).
→ We have added annex A and place a couple of tables which are showing SFOC(Specific fuel oil consumption) and emission factors. Please find annex A of updated manuscript.
With these changes to our final manuscript, we hereby resubmit our manuscript for a secondary evaluation. Thank you once again for your consideration of our paper.
Sincerely,

Round 2
Reviewer 1 Report
In my opinion the manuscript has been significantly
improved and now warrants publication in Sustainability.
Author Response
Dear reviewer,
We would like to express our appreciation to the reviewers for their insightful comments on our paper. We have updated our final manuscript with grammar and spell check. Thank you once again for your consideration of our paper.
Sincerely,

Reviewer 2 Report
We highly appreciate the efforts done to match the suggestions.
Author Response

(The authors gave the same response as above.)

Reviewer 3 Report
All of my remarks have been taken into account and I have no further ones.
Author Response

(The authors gave the same response as above.)
